# Application of ARIMA, and hybrid ARIMA Models in predicting and forecasting tuberculosis incidences among children in Homa Bay and Turkana Counties, Kenya

**Stephen Siamba** *, Argwings Otieno, Julius Koech

University of Eldoret, School of Science, Department of Mathematics and Computer Science, Eldoret, Kenya

* stephen.siamba@gmail.com

**Data Availability Statement:** The authors confirm that the data supporting the findings of this study are available within the article [and/or] as part of

## Abstract

Tuberculosis (TB) infections among children (below 15 years) is a growing concern, particularly in resource-limited settings. However, the TB burden among children is relatively unknown in Kenya where two-thirds of estimated TB cases are undiagnosed annually. Very few studies have used Autoregressive Integrated Moving Average (ARIMA), and hybrid ARIMA models to model infectious diseases globally. We applied ARIMA, and hybrid ARIMA models to predict and forecast TB incidences among children in Homa Bay and Turkana Counties in Kenya. The ARIMA, and hybrid models were used to predict and forecast monthly TB cases reported in the Treatment Information from Basic Unit (TIBU) system by health facilities in Homa Bay and Turkana Counties between 2012 and 2021. The best parsimonious ARIMA model that minimizes errors was selected based on a rolling window cross-validation procedure. The hybrid ARIMA-ANN model produced better predictive and forecast accuracy compared to the Seasonal ARIMA (0,0,1,1,0,1,12) model. Furthermore, using the Diebold-Mariano (DM) test, the predictive accuracy of ARIMA-ANN versus ARIMA (0,0,1,1,0,1,12) model were significantly different, p<0.001, respectively. The forecasts showed a TB incidence of 175 TB cases per 100,000 (161 to 188 TB incidences per 100,000 population) children in Homa Bay and Turkana Counties in 2022. The hybrid (ARIMA-ANN) model produces better predictive and forecast accuracy compared to the single ARIMA model. The findings show evidence that the incidence of TB among children below 15 years in Homa Bay and Turkana Counties is significantly under-reported and is potentially higher than the national average.

## Author summary

Tuberculosis remains a disease of major public health concern especially in resource limited settings. Despite this, tuberculosis is still characterized by high morbidity and mortality from a single infectious disease, particularly among children in developing countries. The actual burden of tuberculosis among children is relatively unknown and about two-

the supporting information (named S1 File) in a comma-separated values format.

**Funding:** The authors received no specific funding for this work.

**Competing interests:** The authors have declared that no competing interests exist.

thirds of cases are either unreported or undiagnosed in Kenya. The use of novel mathematical models is critical and can be leveraged to guide policymakers in the prevention and control of infectious diseases such as tuberculosis. We use autoregressive moving average and hybrid forms of these models to model and forecast tuberculosis infections among children. We found out that hybrid autoregressive moving average models provide more accurate predictions and forecasts of tuberculosis infections among children. We also found out and confirmed that the actual burden of tuberculosis among children is still under-estimated. Our study highlights on the ever existing gap in the under-estimation of tuberculosis among children and points to the importance of novel modelling methods in the understanding of the actual burden of tuberculosis among children.

## Background

Tuberculosis is a highly infectious disease ranked among the top ten most lethal causes of mortality. Approximately 33% of the global population, particularly in developing countries, has been plague-ridden with TB [1]. In 2016, over 10 million new TB cases were reported globally with children below 15 years of age accounting for about 7% of those cases [2]. Furthermore, in 2016, developing countries accounted for over 85% of new TB cases globally with Asian and African countries contributing 61% and 25% respectively of global new TB cases while approximately 7 countries, globally, accounted for close to 65% of all new TB cases [2]. In 2018, about 1 million TB cases and over 230,000 TB-related deaths occurred among children below 15 years with about 55% of these reported TB cases either undiagnosed and/or unreported [3]. In 2019, 30 high TB burdened countries accounted for 87% of all new TB cases while only 8 countries accounted for approximately 67% of the total new TB cases [4]. Despite these statistics, pediatric TB is usually overlooked [5] amid diagnosis and treatment challenges.

The TB burden in Sub-Saharan Africa (SSA) is far much greater and is exacerbated by poverty, political strive, and weak health systems which have curtailed implementation of TB control interventions. Consequently, TB has become an enormous burden to health systems that are already overstretched [6].

Tuberculosis is a disease of major concern in Kenya and is among the top five causes of mortality. Kenya is listed among the top 30 TB high burdened countries [7]. Kenya is also among 14 countries globally that suffer from the TB, TB-Human Immunodeficiency Virus (HIV) and Acquired Immunodeficiency Syndrome (AIDS) co-infection, and Multi-Drug Resistant TB [8] triple burden. The TB incidence for Kenya in 2015 was 233 per 100,000 (95% Confidence Interval (CI): 188–266) population with a mortality of 20 per 100,000 and TB case notification increased from 11,000 to 116,723 between 1990 and 2007 [9] occasioned by the HIV epidemic and improved case detection due to improved diagnostic capacity.

The use of mathematical models in the modeling of epidemic interactions and occurences within populations has been detailed extensively. While existing interventions to control TB have been partially successful, within the context of resource constraints, mathematical modeling can increase understanding and result in better policies toward implementation of effective strategies that would compound better health and economic benefits [10]. In addition, mathematical models such as machine learning (ML) methods are essential and can be leveraged [11] in guiding policymakers in resource allocation toward the prevention and control of diseases.

In Africa, the application of novel machine learning approaches, such as ARIMA models, in modelling disease incidence is well documented. These models, in different forms, have

been used to forecast short-term and long-term patterns of non-infectious diseases such as cancer and malaria [12,13,14]. In these studies, as much as ARIMA models offered a way of predicting cases, they did not guarantee perfect forecasts especially over a longer forecast horizon [12] and can best be applied on data that is stable or exhibits a consistent pattern over time and with minimum outliers [13]. As such, these models would not be suitable if there is no clear strategy of dealing with outliers and suffer from lack of enough data which can result in either under-fitting or over-fitting [14].

More recently, ARIMA and seasonal ARIMA models have been applied to predict and forecast COVID-19 cases in Sub-Saharan Africa. While noting that time series models have been extensively used as convenient methods to predict the prevalence or spreading of infectious diseases, Takele [15] applied ARIMA model to project Covid-19 prevalence in East Africa countries of Ethiopia, Djibouti, Sudan and Somalia. They noted that future prediction of COVID-19 cases especially in the context of the four countries considered in the study might be affected because of the nature of the spread of COVID-19 [15]. In addition, the study did not take into account the effect of seasonality, such as, days of the week where COVID-19 infections were either highest or lowest and this might have impacted on the accuracy of their findings.

Furthermore, Umunna and Olanrewaju [16] modelled HIV prevalence in Minna in Niger state in Nigeria using ARIMA and SARIMA models using monthly HIV data from 2007 to 2018. A SARIMA model was shown to be the best model for forecasting monthly HIV prevalence. Of interest in their findings was that the average fitted value from January 2007 was half of the actual value reported which in essence would indicate under-fitting and might have been better addressed by considering a more robust approach for model evaluation. In addition, outliers which might have accounted for extraneous variation might have been present in the data basing on the 95% prediction intervals which included negative values. Furthermore, the optimal SARIMA model might have been impacted by the existing non-linearities within the data which were not effectively accounted for by the linear model.

In the context of TB, Aryee et al. [17] conducted a study to obtain a time series model to estimate the incidence of TB cases at the chest clinic of the Korle-Bu Teaching hospital (KBTH). They utilized the Box-Jenkins ARIMA approach on monthly TB cases reported at the KBTH from 2008 to 2017. Although they found no evidence of increasing or decreasing trend in the TB incidence, they noted that the best model does not always produce the best results with respect to the mean absolute error (MAE) and mean square error (MSE). As such, the study could have utilized a more robust model and methodology that would further result in better accuracy.

In addition, Ade et al. [18] conducted a study to determine changes in TB epidemiology in last 15 years between 2000 and 2014 in Benin, seasonal variations, and forecasted numbers of TB cases over a period of five years using the Box-Jenkins approach of the ARIMA model. They found existing seasonal variations in TB case finding and notification with the highest numbers recorded within the first quarter of the year. They found that the annual notified cases increased, with the highest reported in 2011 and their 5-year forecast showed a decreasing trend. The study forecasted TB cases over a period of 5 years which would produce inaccurate forecasts because the MSE tends to increase with increase in the forecast horizon. Furthermore, improved accuracy would have been achieved by implementing validation procedures.

Several studies have utilized ARIMA, Seasonal ARIMA (SARIMA), neural network, and hybrid ARIMA models to model TB incidences [19,20] and in these studies, the hybrid models were demonstrated to offer better predictive and forecast accuracy. Azeez et al. [20] compared the predictive capabilities of the SARIMA and the hybrid SARIMA neural network auto-

regression (SARIMA-NNAR) models in modeling TB incidences in South Africa and the SARIMA-NNAR model was found to have better goodness-of-fit. As one of their limitations, Azeez *et al.* [20] noted that the data used covered 2010 to 2015 and were verified against only one year of TB prevalence data and as such, the findings should be interpreted with caution. They proposed that the analysis should be revisited with additional time series data using a strong mathematical model. In this case, availability of data was a gap within this study and as such, more robust approaches in model accuracy improvement would have worked better especially in the context of a small set of data.

Li *et al.* [21] compared the predictive power of the ARIMA and ARIMA-generalized regression neural network (GRNN) hybrid models in forecasting TB incidences in China and concluded that the hybrid model was superior to the single ARIMA model. In this comparative study, as much as the hybrid ARIMA-GRNN hybrid model produced predictions and forecasts with better accuracy, the ARIMA and GRNN single models might have suffered from their inability to effectively account for non-linearities and linearities existing within the data respectively in addition to the lack of enough data to allow better learning from the GRNN model specifically.

The ARIMA, different forms of Neural Networks models and hybrid models have also been applied in modeling other infectious diseases [22,23,24] and in all these studies, hybrid models were found to offer better predictive and forecasting accuracy compared to single models mostly because of their ability to model both linear and non-linear patterns within data.

While hybrid ARIMA models have been applied in forecasting both the short-term and long-term incidences of infectious diseases in other countries, there has been little to no application of these cutting-edge methods in African countries with the majority of the models limited to only ARIMA models. In Kenya, while ARIMA models have been applied in forecasting disease incidence [25], very little has been done in the application of hybrid ARIMA models in predicting disease incidence except in non-public health settings such as agriculture and economics.

The popularity of ARIMA models stems from their flexibility to represent varieties of time series with simplicity but with a profound limitation stemming from their linear assumptions which in many cases is usually impractical [26] since real-world applications mainly involve data exhibiting non-linear patterns. Consequently, to overcome this disadvantage, non-linear stochastic models such as the ANN models have been proposed [27]. Despite this, a single ANN model is not able to incorporate both linear and non-linear patterns and this has led to the adoption of hybrid models to address this challenge [28]. To attain a higher degree of predictive and forecasting accuracy, theoretical and empirical findings show that combining different models can be effective [29].

To better understand the status of TB infection among children in Kenya, it is important to assess the trend and forecast these incidences using available surveillance data and novel models to elicit a better understanding and innovative interventions to curtail the spread of pediatric TB in Kenya. This study compares linear-based ARIMA, and hybrid ARIMA models in modeling TB incidences among children below 15 years in Homa Bay and Turkana Counties in Kenya.

## Materials and methods

### Study design

This was a retrospective quantitative study that utilized aggregated monthly TB cases data reported by health facilities located in Homa Bay and Turkana Counties to the National Tuberculosis, Leprosy and Lung Disease Program (NTLLDP) in the Treatment Information from

Basic Unit (TIBU) electronic system between January 2012 to December 2021 comprising 120 observations of monthly aggregated TB cases for children below 15 years. The study utilized data reported by health facilities in Homa Bay and Turkana Counties which are among the top 10 TB endemic Counties in Kenya [30].

## Study setting

Homa Bay County comprises 8 Sub-Counties and is one of the former districts of Nyanza province in Kenya with Homa Bay town as its headquarter. On the other hand, Turkana County is majorly semi-arid and is made up of 6 Sub-Counties and borders 3 countries of Ethiopia to the North, South Sudan to the North West and Uganda to the West. Homa Bay County is situated on the shores of Lake Victoria, which provides a significant source of income to the local population. Homa Bay County is approximately 3,155 km$^2$ and lies approximately 0.6221˚ S, 34.3310˚ E (S1 Fig) [31]. Turkana County is located 3.3122˚ N, 35.5658˚ E within the former Rift Valley province of Kenya (S1 Fig) [31] and is by far the largest County in Kenya by land area and occupies approximately 68,680 km$^2$ with Lodwar being its largest town and headquarters. Homa Bay and Turkana Counties have a population of approximately 1,131,950 and 926,976 [32] respectively. Homa Bay County has a HIV prevalence that is 4.5 times higher than the national HIV prevalence [33] and faces a double burden of TB-HIV co-infection resulting in an increased risk of TB-related deaths [34]. The population of Turkana is majorly nomadic and is considered a hardship area, prone to drought and faces high disease burden due to inadequate public health resources [35].

## Data collection and analysis

Tuberculosis case data were abstracted and aggregated for each month between January 2012 to December 2021 for health facilities located in Homa Bay and Turkana Counties in Kenya. In 2012, the Kenya Ministry of Health (MoH) through the Division of Leprosy, Tuberculosis and Lung Disease transitioned the reporting of TB cases from paper-based to the TIBU system [36]. The TIBU system is a national TB case-based surveillance system used in the storage of individual cases of TB that are reported to the national TB program monthly with nationwide coverage [37]. This study did not collect or utilize patient-level data.

One of the objectives of time series analysis is to use an observed time series to forecast future observations. In the absence of actual new data to forecast, the cross-validation technique offers a way through which a model's future predictive accuracy is estimated and errors minimized. In addition, Arlot and Celisse [38], noted that given that training a model and evaluating its performance on the same data results in overfitting and because of working with limited data, splitting the data into a training and validation sample suffices. In the context of cross-validation, while a single data split yields a validation estimation of risk, averaging over a number of splits yields a cross-validation estimate. In this case, a minimum size for the training set was specified and based on one-step forecasts [39], different training sets, each containing one more observation from the previous one were used [40].

Data analysis was performed using R statistical software [41] together with applicable packages for analyzing time-series data. The results were summarized using tables and figures.

## The Time Series concept

A time series is a sequential set of data measured over time and is typically composed of the trend, cyclical, seasonal, and irregular (random) components.

An autoregressive (AR) model is a type of random process used to describe certain time-varying processes within a time series [42]. The basic idea of AR models is that the present

value of a series $Y_t$ can be linearly explained by a function of $p$ past values, that is, $Y_{t-1}$, $Y_{t-2}$,...,$+Y_{t-p}$.

In the case of this study, the expected value of the series $Y_t$ was not equal to zero (0), that is, $E(Y_t) = \mu \neq 0$, as such, the series $E(Y_t) = \mu \neq 0$, $Y_t$ was replaced by $Y_t - \mu$ to obtain an AR process of order $p$ [42] and can be written as.

$$Y_t = \alpha + \phi_1 Y_{t-1} + \phi_2 Y_{t-2} + \ldots + \phi_p Y_{t-p} + \varepsilon_t \tag{1}$$

Where; $\varepsilon_t$ is white noise (WN), is uncorrelated with $Y_s$ for all $s < t$ and $\alpha = \mu(1-\phi_1-\ldots-\phi_p)$

A moving average (MA) model uses the dependency between an observed value and the residual error from a moving average model applied to lagged observations. This implies that the output variable is linearly dependent on the current and past values of a stochastic term [42].

Consequently, $Y_t$ is a moving average process of order $q$ if;

$$Y_t = \varepsilon_t + \theta_1 \varepsilon_{t-1} + \theta_2 \varepsilon_{t-2} + \ldots + \theta_q \varepsilon_{t-q}, \tag{2}$$

Where $\varepsilon_t$ is WN and $\theta_1$,...,$\theta_q$ are constants

Alternatively, Eq 2 can also be written in the form $Y_t = \theta(B)\varepsilon_t$, where $\theta(B) = 1 + \theta_1 B + \theta_2 B^2 + \ldots + \theta_q B^q = 1 + \sum_{j=1}^{q} \theta_j B^j$ is the moving average operator.

## Autoregressive Integrated Moving Average (ARIMA) models

A non-seasonal ARIMA ($p$, $d$, $q$) model is a class of stochastic processes whose auto-covariance functions depend on a finite number of unknown parameters. The ARIMA model can only be applied when a series is stationary [43] which can be achieved by differencing the series. Generally, an ARIMA process of orders $p$, $d$ and $q$ can be represented mathematically [44] as;

$$Y_t = \mu + \sum_{j=1}^{p} \phi_j Y_{t-j} + \sum_{j=1}^{q} \theta_j \varepsilon_{t-j} + \varepsilon_t \forall t \in \mathbb{Z} \tag{3}$$

In lag operator notation, a non-seasonal non-differenced ARIMA ($p$, $d$, $q$) process is written as $\phi(B)Y_t = \theta(B)\varepsilon_t$ $\forall t \in \mathbb{Z}$.

Box and Jenkins introduced the ARIMA model in 1960 [45]. The ARIMA model requires only historical time series data on the variable under forecasting. Most importantly, ARIMA models are represented as ARIMA ($p$, $d$, $q$) where $p$ is the number of AR terms, $d$ is the number of non-seasonal differences, and $q$ is the number of lagged forecast errors [46]. The ARIMA model assumes that the residuals are independent and normally distributed with $\varepsilon_t \sim N(\mu, \sigma^2)$ homogeneity of variance and zero mean value.

## Seasonal Autoregressive Integrated Moving Average models (SARIMA) models

The SARIMA model is made up of non-seasonal and seasonal components in a multiplicative model. A SARIMA model can be written as ARIMA ($p$,$d$,$q$) ($P$,$D$,$Q$)$^S$ where $p$ is the non-seasonal AR order, $d$ is the non-seasonal differencing, $q$ is the non-seasonal MA order, $P$ is the seasonal AR order, $D$ is the seasonal differencing, $Q$ is the seasonal MA order and $S$ is the period of repeating seasonal pattern. Generally, $S = 12$ for monthly data. As such, with the backshift operator presented as $BY_t = Y_{t-1}$, without differencing, a SARIMA model can be written formally as [47];

$$\vartheta(B^s)\phi(B)(Y_t - \mu) = \theta(B)\Theta(B^s)\varepsilon_t, \forall t \in \mathbb{Z} \tag{4}$$

Where on the left of Eq 4, the seasonal and non-seasonal AR processes multiply each other, and on the right, the seasonal and non-seasonal MA processes multiply each other. Also, in this study, S = 12 since monthly TB cases was used.

## Artificial Neural Networks (ANNs) models

Artificial Neural Networks have been suggested as alternative and better modeling approaches to time series forecasting [48]. The main goal of ANNs is to construct a model that mimics the human brain intelligence into a machine [47,48,49] and are biologically motivated [49]. The most common ANNs are multi-layer perceptrons (MLPs) [50] made up of the input layer, the hidden layer, and the output layer connected by acyclic links [51]. A neuron is a data processing unit while the nodes in the various layers of ANNs are the processing elements.

The ANN model equation can be presented according to Zhang [52] and it performs a nonlinear mapping from past observations of a time series to a future value. In addition, there is no systematic rule in deciding the choice of $q$ while $p$, which is the number of neurons, is equal to the number of features in the data [52]. The logistic function h(.) is applied as the nonlinear activation function represented as, $h(.) = \frac{1}{1+e^{-x}}$.

## Hybrid (ARIMA-ANN) models

Generally, a time series can be observed as having linear and nonlinear components as $Y_t = l_t + n_t$ Where $l_t$ and $n_t$ are the linear (from the ARIMA model) and nonlinear (ANN fitted ARIMA model residuals) components respectively. Residuals from the ARIMA model are fitted with the ANN model.

## Proposed methodology

The proposed methodology for this study was based on the combination of the Box-Jenkins methodology for ARIMA modeling, and the hybrid ARIMA models. First, the ARIMA model was developed with the optimal model selected based on the minimum AIC and BIC as well as the model that minimizes RMSE, MAE, and MAPE. This was achievable by applying an automated ARIMA function following Box and Jenkins procedure [52] within a cross-validation procedure. Second, the best parsimonious ARIMA model was used to predict TB cases and accuracy measures calculated by comparing the fitted and the actual TB cases. Later, the model was used to forecast TB cases for the year 2022. Third, the residuals from the ARIMA model were obtained and fit using an autoregressive neural network to ensure that any existing signal was captured. Fourth, the fitted residuals were combined with the ARIMA model fitted TB cases to form the hybrid model. The fitted TB cases of the hybrid model were compared against the actual TB cases and accuracy measures calculated. Fifth, the hybrid model was used to forecast TB cases for the year 2022. Finally, the predictive accuracy of the two models was compared to establish the model with the best predictive accuracy.

## Model identification and specification

Optimal values of $p$, $d$, $q$, $P$, $D$, and $Q$ for the ARIMA model were determined by examining the autocorrelation functions, and the best model was determined by testing models with different parameters of $p$, $d$, $q$, $P$, $D$, and $Q$. The models were estimated using the maximum likelihood estimation (MLE) method and the Akaike Information Criterion (AIC) and Bayesian Information Criterion (BIC) [53] penalty function statistics were used to determine the best model that minimizes AIC or BIC.

One assumption of the ARIMA model is that the residuals should be white noise. As such, the Ljung-Box Q test [54] was used to test the hypothesis of independence, constant variance and zero mean of the model residuals.

## Accuracy measures

Various accuracy measures have been proposed [55] to determine predictive and forecast performance. This study used the Root Mean Squared Error (RMSE), Mean Absolute Error (MAE), and the Mean Absolute Percent Error (MAPE) to measure the predictive and forecast accuracy of the two models. The lower the values of these accuracy measures the better the model. Furthermore, MAPE values of 10% or below, 10–20%, and 20–50% should be considered as high accuracy, good accuracy, and reasonable accuracy [56].

The study also compared the predictive accuracy of the forecasts from the three models using the Diebold-Mariano (DM) test [57]. The test was used to test the null hypothesis that two models have similar predictive accuracy.

To allow implementation of the cross-validation procedure, the minimum number of observations required to fit the ARIMA model was set basing on the recommendation by Hyndman and Kostenko [58] who proposed that at least $p+q+P+Q+d+mD+1$ observations are sufficient for a seasonal ARIMA model in which case, the study considered a seasonal ARIMA model, though automatically selected basing on the fact that the data used was monthly data and seasonality had to be accounted for. In addition, the minimum number of observations for model development within the cross-validation framework was set at 60, comprising of observations for 5 years [59].

## Ethical approval and considerations

A research permit was obtained from the National Commission for Science, Technology, and Innovation (NACOSTI) in Kenya. Authorization for use of the data from the TIBU system was obtained through an letter of approval under the Patient and Program Outcomes Protocol (PPOP) by the Elizabeth Glaser Pediatric AIDS Foundation.

# Results

## Exploratory data analysis

There was a total of 120 observations in this data. The trend of the TB cases among children below 15 years in Homa Bay and Turkana Counties in the data (Fig 1) showing a notable increase in the TB cases reported between 2018 and 2021. The monthly cycle box plot of TB cases (S2 Fig) show that there is a potential presence of seasonality within the reported TB cases. However, whether or not to account for seasonality in the model depends on whether this would improve model accuracy. This implies that there is need to account for seasonality within the ARIMA model. Furthermore, outliers were detected in some months.

## Comparison of model performance in predicting TB cases

**Model estimation and accuracy.** The Akaike Information Criterion (AIC) and the Bayesian Information Criterion (BIC) were used to pick the best parsimonious model based on the least AIC or BIC estimated values. The best model was ARIMA (0,0,1,1,0,1,12); where $p = 0$, $d = 0$ and $q = 1$ respectively and $P = 1$, $D = 0$ and $Q = 1$ respectively. The Ljung-Box Q test for the best model showed a p-value of 0.079 implying that the ARIMA (0,0,1,1,0,1,12) model residuals were independently distributed. The best model was made up of non-differenced seasonal AR (1), non-seasonal MA (1) model and seasonal MA (1) polynomials. From the model

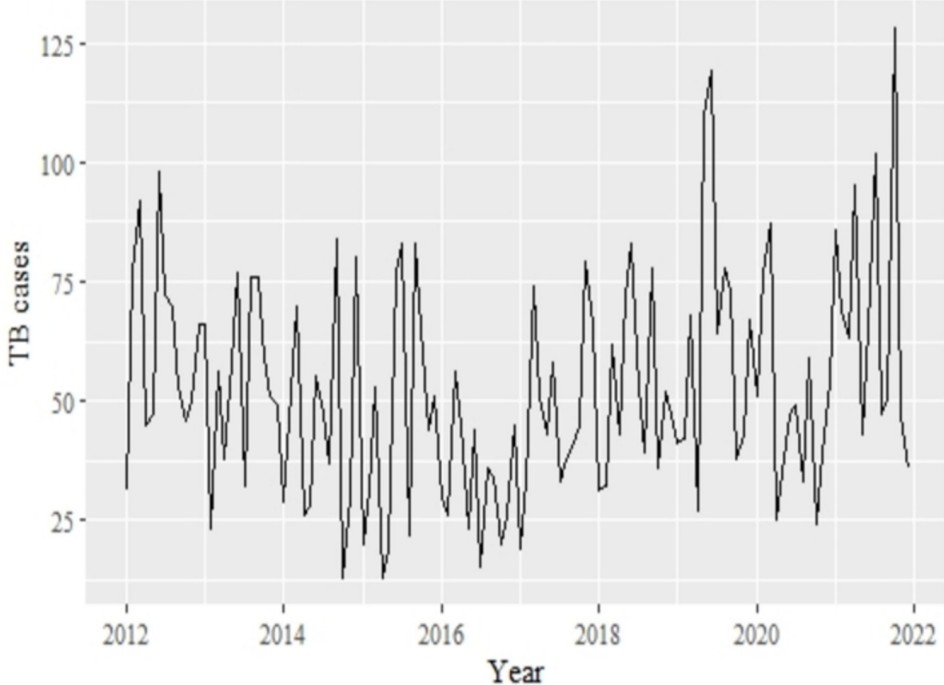

**Fig 1. Monthly TB cases among children below 15 years from Homa Bay and Turkana Counties between 2012 and 2021.**

output, the estimated coefficients were (Table 1); ma1 = $\theta_1$ = 0.296, sar1 = $\vartheta_1$ = 0.999, sma1 = $\Theta_1$ = -0.968 and $\mu$ = 50.698.

Plugging these estimated coefficients into Eq 4 yields the model equation:

$$(1 - 0.999B^{12})(Y_t - 50.698) = (1 - 0.968B^{12})(1 + 0.296B)\varepsilon_t, \forall t \in \mathbb{Z} \qquad (5)$$

**Seasonal ARIMA model diagnostics and performance.**   The performance of the Seasonal ARIMA (0,0,1,1,0,1,12) model was carried out by comparing predicted and forecasted TB cases with the actual TB cases reported (Fig 2).

Comparison of the accuracy of parameters/measures of the Seasonal ARIMA (0,0,1,1,0,1,12) model fitted against the actual TB cases showed a RMSE, MAE and MAPE values of 18.69, 14.32, and 38.93 respectively. In addition, the mean number of fitted TB cases from the Seasonal ARIMA (0,0,1,1,0,1,12) model was 51 cases compared to a mean of 51 cases

**Table 1. Estimated SARIMA model coefficients.**

| Parameters | Coefficients | Std. Error | Z-value | Pr(>|z|) |
|---|---|---|---|---|
| ma1 | 0.296 | 0.108 | 2.743 | 0.006* |
| sar1 | 0.999 | 0.007 | 143.639 | <0.001** |
| sma1 | -0.968 | 0.088 | -10.975 | <0.001** |
| Intercept | 50.698 | 5.070 | 9.999 | <0.001** |

Significance codes: 0 '***' 0.001 '**' 0.01 '*' 0.05 '.' 0.1 ' ' 1

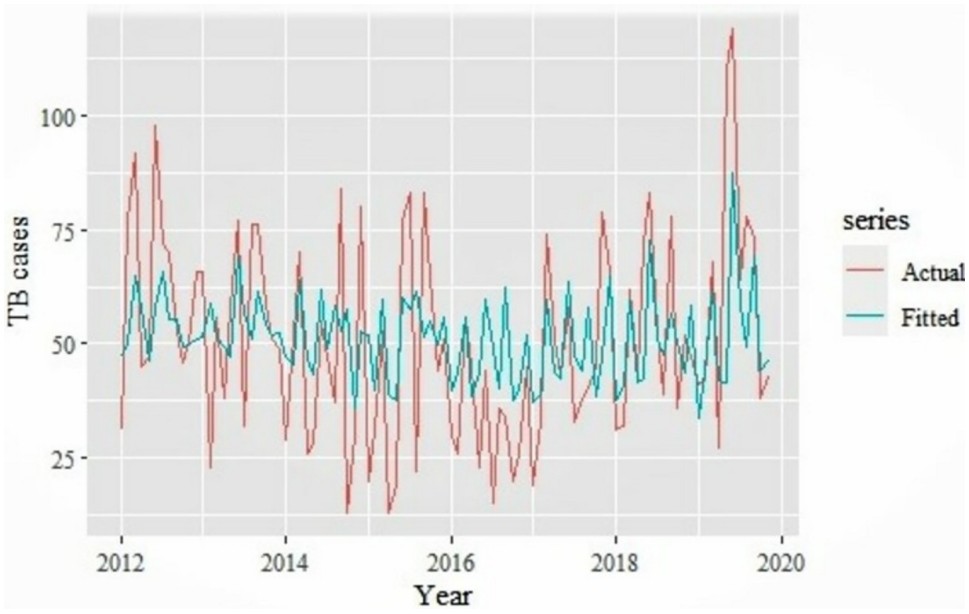

**Fig 2. Comparison of Seasonal ARIMA fitted versus actual TB cases.**

from the actual reported TB cases. The monthly median plots with the fitted and actual median TB cases compared, and it clearly shows that the model is able to capture the seasonal pattern within the fitted TB cases as well (Fig 3).

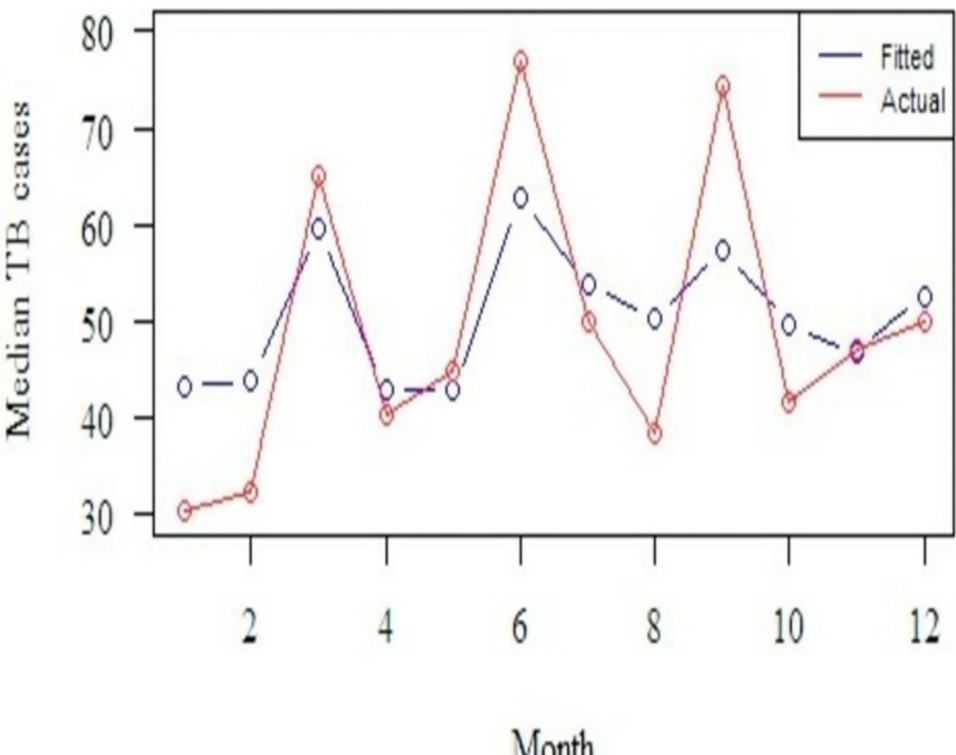

**Fig 3. Monthly median plots for fitted versus actual TB cases.**

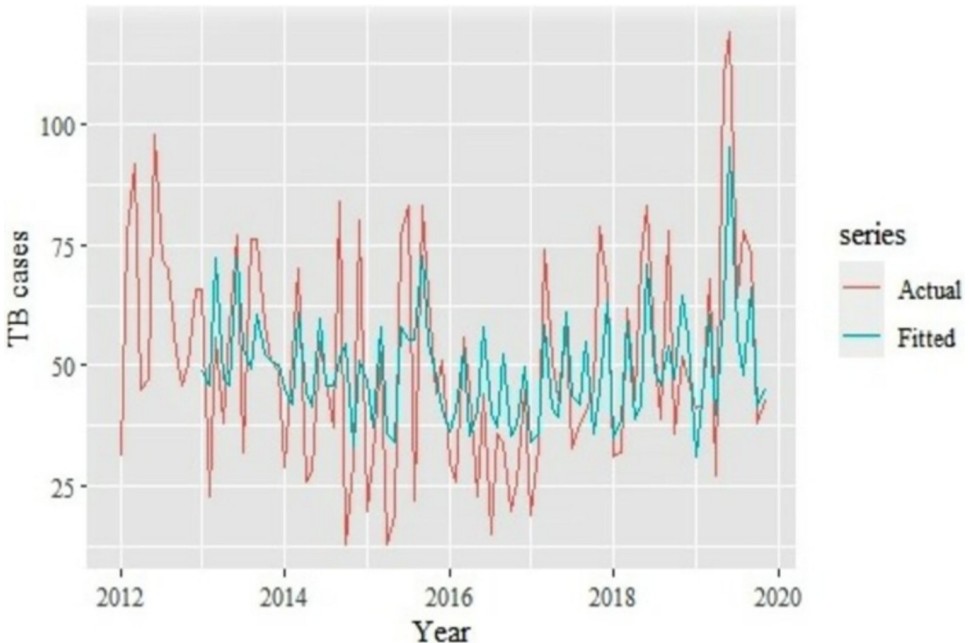

**Fig 4. Comparion of seasonal ARIMA-ANN predicted TB cases versus actual TB cases.**

The best ARIMA (SARIMA) model was assessed for fit using the standard model residual analysis (S3 Fig). The residuals plot was relatively normal except for a few outliers at the tails, with model residuals being normally distributed. Inspection of the Autocorrelation Function (ACF) test residual randomness in order to identify patterns or extreme values showed significant auto-correlations at lag 3.

**Hybrid (Seasonal ARIMA-ANN) model estimation and accuracy.** Residuals from the optimal Seasonal ARIMA (0,0,1,1,0,1,12) model were fit using an ANN model and the accuracy measures calculated as well as comparison of the forecast and prediction (Fig 4). The residuals from the optimal Seasonal ARIMA (0,0,1,1,0,1,12) model were fit using an ANN model using the Neural Network Auto-Regressive (NNAR) function to produce an NNAR ($p$, $P$,$k$) [m] model. The optimal lag parameter, $p$, and the number of nodes in the hidden layer, $k$, were automatically selected while $P = 1$ by default. In addition, a decay parameter of 0.001 and a maximum iteration of 200 were pre-set for the model to help restrict the weights from becoming too large and ensure that the model can test different models until the optimal model that has the minimal RMSE produced respectively.

The findings show that the hybrid Seasonal ARIMA-ANN model resulted in a RMSE, MAE, MAPE values of 16.41, 12.99, and 36.00 respectively when the fitted TB cases from the hybrid model were compared against the actual TB cases reported. This represents a decrease of 12.2%, 9.3%, and 7.5% on the RMSE, MAE, and MAPE accuracy measures respectively when compared to the accuracy of the Seasonal ARIMA model.

**Comparison of model predictive accuracy.** The predictive accuracy of the models was compared using the Diebold-Mariano (DM) test with the null hypothesis that the predictive accuracy of the two models compared are the same. The DM statistic was 3.819, with a p-value of <0.001 indicating that the Seasonal ARIMA-ANN and Seasonal ARIMA (0,0,1,1,0,1,12) models present significantly different predictive accuracies. In general, the Seasonal ARIMA-ANN model offers better predictive accuracy compared to the Seasonal ARIMA (0,0,1,1,0,1,12) model.

**Table 2. Tuberculosis cases 2022 point forecasts.**

| Month | Seasonal ARIMA (0,0,1,1,0,1,12) | Seasonal ARIMA-ANN |
|---|---|---|
| Jan-22 | 39 | 44 |
| Feb-22 | 47 | 43 |
| Mar-22 | 65 | 62 |
| Apr-22 | 43 | 43 |
| May-22 | 49 | 48 |
| Jun-22 | 68 | 72 |
| Jul-22 | 54 | 52 |
| Aug-22 | 48 | 49 |
| Sep-22 | 61 | 62 |
| Oct-22 | 48 | 49 |
| Nov-22 | 47 | 50 |
| **Mean** | **52** | **52** |
| **Total** | **569** | **573** |

**Comparison of model performance in forecasting temporal trends of TB incidences.**
The resulting Seasonal ARIMA (0,0,1,1,0,1,12), and ARIMA-ANN models were used to forecast TB cases for 2022. The point forecast results (Table 2) and comparison of the model forecasts (Fig 5) show the mean forecasted TB cases was 52 (80% CI: 48, 56), and 52 (80% CI: 48, 56) cases per month based on the Seasonal ARIMA (0,0,1,1,0,1,12), and hybrid ARIMA-ANN respectively for 2022 (upto November) giving a total of 569, and 573 TB cases forecasted for the year 2022 (upto November) from the Seasonal ARIMA (0,0,1,1,0,1,12), and ARIMA-ANN models respectively.

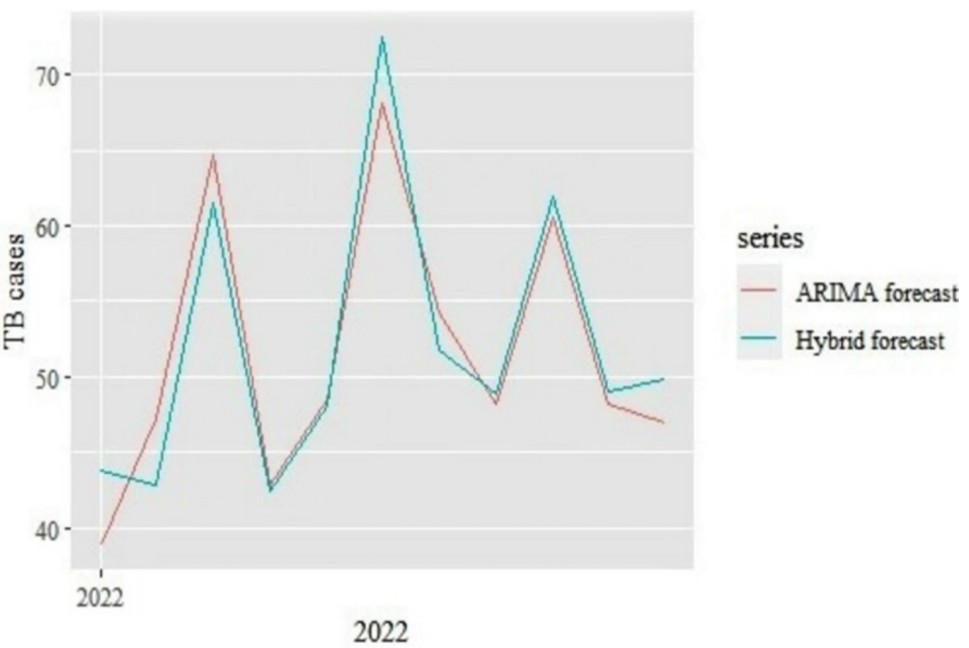

**Fig 5. Forecasted TB cases for 2022.**

## Discussion

Although the two models were able to predict TB cases among children below 15 years, the hybrid Seasonal ARIMA-ANN model was able to offer better predictive performance compared to the single Seasonal ARIMA model. These findings compare with those from other studies which applied either hybridized ARIMA or SARIMA in the modeling of TB incidences and other infectious diseases [19,23,59,60] with the overall conclusion that hybrid models have better predictive performance. The majority of infectious disease data are neither purely linear nor non-linear and mostly present with both linear and nonlinear properties. As such, single models are not enough in modeling such kinds of data. Hybrid models are found to be most appropriate for the accurate estimation of such data [61]. The use of hybridized ARIMA models has been proposed in recent years and used extensively with improvements proposed over time.

The estimated TB incidence in Kenya was 259 TB cases per 100,000 population in 2020 [5] in the general population. This translates to approximately 134,680 TB cases, and with children accounting for about 20% (26,936) of these cases [62], the incidence among children below 15 years was approximately 121 TB cases per 100,000 population of children. Furthermore, children present as a most vulnerable with a higher risk of contracting TB [63]. In addition, Makori *et al.* [30] noted that the burden of TB in Kenya was higher than previously thought. This study forecasted a mean of 52 TB cases per month in 2022 (till November) for Homa Bay and Turkana Counties and estimates that the mean number of TB cases reported among children below 15 years would be approximately 624 in 2022. However, given that these are estimated reported cases, they most likely represent only about 35% of TB cases since up to 65% of pediatric TB cases are potentially missed each year [3]. Taking this into account, the estimated TB cases for 2022 will be approximately 1783 and ranging between 1646 to 1920 for Homa Bay and Turkana Counties among children below 15 years on average. The estimated population of children below 15 years in Homa Bay and Turkana Counties for 2022 is approximately 1,020,795 [64]. As such, the estimated TB incidence among children in Homa Bay and Turkana Counties in 2022 would be approximately 175 TB incidences per 100,000 population (161 to 188 TB cases per 100,000 population). This estimated TB incidence among children below 15 years for Homa Bay and Turkana Counties is slightly lower compared to the estimated TB incidence in 2015 which was estimated at 233 TB cases per 100,000 (95% CI 188–266) population within the general population in Kenya [6] but higher than the estimated national average of 121 TB cases per 100,000 population of children below 15 years in 2020.

The findings of this study show that the estimated TB incidence among children below 15 years is higher compared to the estimated national average for 2020. These findings are in line with the WHO newsletter that indicated that the number of people developing TB and dying from the disease could be much higher in 2021 and 2022 mainly because of the COVID-19 pandemic [65] and since this was based on the general population, it is concerning that the same trend is witnessed among children below 15 years. These findings also confirm those by Oliwa *et al.* [66] who indicated that notification data may underestimate the TB burden among children while Mbithi *et al.* [67] reported a decrease in TB diagnosis in Kenya by an average of 28% in the year 2020. In addition, the conclusions from Makori *et al.* [30] about the need to intensify TB case finding among younger, especially pediatric populations affirm the findings of this study.

The findings of this study further reveal that TB infections among children tended to exhibit a seasonal pattern with 3 peaks experienced in March, June and September respectively. Despite very few studies highlighting the importance of seasonal variations coinciding with TB infections, seasonality of TB infections has been documented in other studies

[68,69,70]. While other studies did not directly attribute TB infections to seasonal patterns, Jaganath *et al.* [71] found a link between the peaks of the rain and influenza seasons and increased TB infections among children in Uganda. The findings in this study showing that TB infections among children correlate with seasons might be due to the fact that different seasonal patterns such as dry and rainy seasons carry a major influence on TB transmission and health seeking behavior within the study area. Prior studies have suggested higher TB infections in rainy seasons which are coupled with higher incidence of respiratory illnesses and lower vitamin D levels [72] and would require further investigation with the aim of putting in specific interventions that would result in increased TB screening and diagnosis in peak seasons and to curtail TB infections in such seasons.

## Conclusion

The hybrid ARIMA model offers better predictive accuracy and forecast performance compared to the single ARIMA model in modeling TB cases among children below 15 years in Homa Bay and Turkana Counties.

The findings in this study confirm that the under-reporting of TB cases among children below 15 years and the incidence in this vulnerable group is still persistent and might be higher than previously estimated. As such, there is need to re-look at the TB surveillance framework data more closely to understand existing gaps. There is an urgency to re-align vital resources towards the National TB program to have the TB fight back on track in these two Counties, especially active case finding among children which would also require application of novel methods of TB diagnosis.

Furthermore, the findings of this study point to the fact that TB infections among children below 15 years in Homa Bay and Turkana Counties are influenced by seasonal patterns which might influence the health seeking behavior and transmission pattern of the disease. As such, there is need to invest resources toward increased TB surveillance, screening, and diagnosis efforts within specific months of the year as well as putting in measures to curtail spread of the disease during peak seasons.

## Limitations

This study utilized data collected and reported in the TIBU system, as such, the study did not have control over the quality and accuracy of the data. However, it was assumed that given that the data had been reported in the system, all related procedures to assure data quality had been followed by the reporting health facilities within Homa Bay and Turkana Counties.

This study utilized data between 2012 to 2021 which comprised 120 observations representing monthly aggregated TB cases for children below 15 years. Deep learning and machine learning algorithms usually demand a large amount of data to allow the algorithm to effectively learn. As such, the available data might not have been sufficient to allow for better learning by the algorithm. As such, the models can be applied with additional data.

This study combined data and analysis for Turkana and Homa Bay County. However, these two Counties might present different scenarios when it comes to pediatric TB. In addition, since the study focused on modeling TB cases among children below 15 years in Homa Bay and Turkana Counties, the findings might not be generalized to other Counties of Kenya.

The study data covered the period 2012 to 2021 which included the years 2020 and 2021 during which the COVID-19 pandemic was experienced in Kenya and the region. During this period, there were COVID-19 related measures and restrictions put in place by the government of Kenya aimed at reducing the spread of the corona virus. As such, such measures would have had an unprecedented effect on TB related activities at community and health

facility level. Consequently, this study could not quantify the COVID-19 impact on TB cases reported among children below 15 years as this was beyond the scope of this study. A possible recommendation is to utilize models such as interrupted time series to measure possible impact of COVID-19 on TB detection, diagnosis and management.

## Supporting information

**S1 Fig. Map of Kenya showing Homa Bay and Turkana Counties.** (https://kenya. africageoportal.com/datasets/d2f2df2a08ef42e88cb6bdc00e41dcc9_0/explore?location=0. 361948%2C41.711735%2C6.00) [31].
(TIF)

**S2 Fig. Monthly cycle plot of TB cases among children below 15 years in Homa Bay and Turkana Counties.**
(TIF)

**S3 Fig. Seasonal ARIMA Model residual diagnostics.**
(TIF)

**S1 File. Tuberculosis Data for children below 15 years between 2012 and 2021 for Homa Bay and Turkana Counties, Kenya.**
(CSV)

## Acknowledgments

We would like to acknowledge the departments of health in Homa Bay and Turkana County and the health facilities within these Counties for the program interventions towards TB identification and management of TB cases. Their efforts have gone a long way in contributing to the data used in this study. We also acknowledge the Elizabeth Glaser Pediatric AIDS foundation for granting permission to use this data within their Patient and Program Outcomes Protocol (PPOP); this made our access process easy while also meeting the ethical requirements.

## Author Contributions

**Conceptualization:** Stephen Siamba.

**Data curation:** Stephen Siamba.

**Formal analysis:** Stephen Siamba.

**Funding acquisition:** Stephen Siamba.

**Investigation:** Stephen Siamba.

**Methodology:** Stephen Siamba.

**Project administration:** Stephen Siamba.

**Resources:** Stephen Siamba.

**Software:** Stephen Siamba.

**Supervision:** Argwings Otieno, Julius Koech.

**Validation:** Stephen Siamba, Argwings Otieno, Julius Koech.

**Visualization:** Stephen Siamba.

**Writing – original draft:** Stephen Siamba.

**Writing – review & editing:** Argwings Otieno, Julius Koech.

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
