## [Decision Letter · Decision Letter 0]

6 Sep 2022

PDIG-D-22-00198

Application of ARIMA, hybrid ARIMA and Artificial Neural Network Models in predicting and forecasting tuberculosis incidences among children in Homa Bay and Turkana Counties, Kenya

PLOS Digital Health

Dear Dr. Siamba,

Thank you for submitting your manuscript to PLOS Digital Health. After careful consideration, we feel that it has merit but does not fully meet PLOS Digital Health's publication criteria as it currently stands. Therefore, we invite you to submit a revised version of the manuscript that addresses the points raised during the review process.

Please submit your revised manuscript within 60 days Nov 05 2022 11:59PM. If you will need more time than this to complete your revisions, please reply to this message or contact the journal office at digitalhealth@plos.org. Please include the following items when submitting your revised manuscript:

We look forward to receiving your revised manuscript.

Kind regards,

Thomas Schmidt

Academic Editor

PLOS Digital Health

Journal Requirements:

Additional Editor Comments (if provided):

Thank you for submitting your work to PLOS Digital Health. Your paper has been submitted to two reviews, and based on their feedback, as well as my own assessment. I recommend a major revision before final acceptance.

I agree with both our reviewers comments, but would also like to add a few of my own. Foremost regarding the structure and content of the paper. I’m of the opinion that research papers should limit the elaborate use of formulas for fairly generic algorithms. They make sense when adjustments or customizations have been utilized. So please consider if the many equations listed in Materials and methods section are truly necessary, or can be found in textbooks. Likewise, I recommend a reconsideration of the value of listing metrics for models during both training and testing. These metrics are useful when doing an overall evaluation of each model’s performance, but consider if they are relevant for potential readers.

I find your paper to be well written but would like to have you elaborate the discussion a bit further on the applicability of your approach.

Also, to what extent is your dataset affected by COVID-19? Please consider this in your limitations section.

Finally, I suggest that you use consistent terminology for your dataset. ‘Data points’ are confusing, please use subjects, patients, or children instead. You use both data points, records etc.

Reviewers' comments:

Reviewer's Responses to Questions

**Comments to the Author**

1. Does this manuscript meet PLOS Digital Health’s publication criteria? Is the manuscript technically sound, and do the data support the conclusions? The manuscript must describe methodologically and ethically rigorous research with conclusions that are appropriately drawn based on the data presented.

Reviewer #1: Partly

Reviewer #2: Partly

2. Has the statistical analysis been performed appropriately and rigorously?

Reviewer #1: Yes

Reviewer #2: Yes

3. Have the authors made all data underlying the findings in their manuscript fully available (please refer to the Data Availability Statement at the start of the manuscript PDF file)?

Reviewer #1: No

Reviewer #2: Yes

4. Is the manuscript presented in an intelligible fashion and written in standard English?

Reviewer #1: Yes

Reviewer #2: Yes

5. Review Comments to the Author

Reviewer #1: General comments 

This study applied ARIMA, hybrid ARIMA and ANN to predict incidence of TB in Children in two counties in Kenya and demonstrates that hybrid ARIMA has better predictive and forecast accuracy in comparison to ARIMA and ANN models. This is an interesting study and including more details in methods would improve the manuscript further. 

Specific comments 

Major comments 

The authors haven’t provided adequate justification for this study apart from stating that previous studies have used only ARIMA models in forecasting disease incidence in Kenya. It would be helpful if the authors conducted a more rigorous literature review to identify the gaps in literature on how ARIMA and hybrid ARIMA models were applied for forecasting disease incidence in Africa and Kenya. Also, the authors must highlight the gaps in Azeez et al.’s study and the methodological gaps in that study to justify the aims and objectives of this study. 

My biggest concern with this study is the selection of ANN models as a comparator. With the limited size of the dataset, these models were always likely to underperform. Adequate justification on why these models have been used for this study and why it is important to compare ANN with ARIMA and hybrid ARIMA would be helpful. Also, multiple other studies have demonstrated that SARIMA 

I would strongly encourage the authors to use Jandoc et al.’s study (https://pubmed.ncbi.nlm.nih.gov/25890805/) and Schaffer et al.’s article (https://bmcmedresmethodol.biomedcentral.com/articles/10.1186/s12874-021-01235-8 ) to report the methodological and reporting recommendations. 

To test the ARIMA, hybrid ARIMA and ANN, it is not clear if the authors conducted a sensitivity analysis. 

Minor comments 

In line 204, did the authors mean ARMA (Autoregressive Moving Average) model or ARIMA model? If it is the former, please expand ARMA. 

I would also encourage the authors to include the data as supplementary material.

Reviewer #2: The authors developed ARIMA, Hybrid ARIMA and ANN models to predict and forecast the incidence of Tuberculosis among under 15 children. The topic is interesting, primarily focusing on paediatric Tuberculosis.

1. Since the number of data points is significantly less (120) and from the methodology, the authors used an 80/20 split for training and testing, it is unclear whether the ARIMA or hybrid ARIMA orders are determined after splitting the training and test set or using the whole dataset. If authors used the entire dataset for determining the order, they could introduce the bias in the test set. What are the measures taken to mitigate the overfitting issues?

2. It is unclear whether the authors used any cross-validation techniques in this work. Many applicable packages are available in R for utilizing the cross-validation for better performance in predicting and forecasting, especially when the data points are less.

3. Model (ARIMA (0,0,1,1,0,1,12)), Model (NNAR (1,1,2) [12]), and Hybrid ARIMA-ANN shows the test errors are higher than training errors. Even though it is common in models, the large variation indicates the overfitting of the model. Can the authors explain the measures taken to reduce the test errors?

6. PLOS authors have the option to publish the peer review history of their article (what does this mean?). If published, this will include your full peer review and any attached files.

**Do you want your identity to be public for this peer review?** For information about this choice, including consent withdrawal, please see our Privacy Policy.

Reviewer #1: No

Reviewer #2: No

---

## [Decision Letter · Decision Letter 1]

15 Dec 2022

Application of ARIMA, and hybrid ARIMA Models in predicting and forecasting tuberculosis incidences among children in Homa Bay and Turkana Counties, Kenya

PDIG-D-22-00198R1

Dear Mr. Siamba,

We are pleased to inform you that your manuscript 'Application of ARIMA, and hybrid ARIMA Models in predicting and forecasting tuberculosis incidences among children in Homa Bay and Turkana Counties, Kenya' has been provisionally accepted for publication in PLOS Digital Health.

Best regards,

Thomas Schmidt

Academic Editor

PLOS Digital Health

Dear Stephen Siamba et al.

Thank you for submitting your revised manuscript to PLOS Digital Health. Sorry about the prolonged processing time. However, as evident from the reviewers final comments, all concerns have been thoroughly and properly dealt with. Thank you. I recommend that your submission be accepted for publication.

Reviewer Comments (if any, and for reference):

Reviewer's Responses to Questions

**Comments to the Author**

1. If the authors have adequately addressed your comments raised in a previous round of review and you feel that this manuscript is now acceptable for publication, you may indicate that here to bypass the “Comments to the Author” section, enter your conflict of interest statement in the “Confidential to Editor” section, and submit your "Accept" recommendation.

Reviewer #1: All comments have been addressed

Reviewer #2: All comments have been addressed

2. Does this manuscript meet PLOS Digital Health’s publication criteria? Is the manuscript technically sound, and do the data support the conclusions? The manuscript must describe methodologically and ethically rigorous research with conclusions that are appropriately drawn based on the data presented.

Reviewer #1: Yes

Reviewer #2: Yes

3. Has the statistical analysis been performed appropriately and rigorously?

Reviewer #1: Yes

Reviewer #2: Yes

4. Have the authors made all data underlying the findings in their manuscript fully available (please refer to the Data Availability Statement at the start of the manuscript PDF file)?

Reviewer #1: Yes

Reviewer #2: Yes

5. Is the manuscript presented in an intelligible fashion and written in standard English?

Reviewer #1: Yes

Reviewer #2: Yes

6. Review Comments to the Author

Reviewer #1: Thank you for thoughtfully responding to my comments and for the excellent paper.

Reviewer #2: (No Response)

7. PLOS authors have the option to publish the peer review history of their article (what does this mean?). If published, this will include your full peer review and any attached files.

**Do you want your identity to be public for this peer review?** For information about this choice, including consent withdrawal, please see our Privacy Policy.

Reviewer #1: No

Reviewer #2: No
